# Role of Inflammation in Pathophysiology of Colonic Disease: An Update

**DOI:** 10.3390/ijms21134748

**Published:** 2020-07-03

**Authors:** Noha Ahmed Nasef, Sunali Mehta

**Affiliations:** 1Riddet Institute, Massey University, Private Bag 11222, Palmerston North 4442, New Zealand; N.Nasef@massey.ac.nz; 2Department of Pathology, Dunedin School of Medicine, University of Otago, Dunedin 9054, New Zealand; 3Maurice Wilkins Centre for Biodiscovery, University of Otago, Dunedin 9054, New Zealand

**Keywords:** inflammation, diverticular disease, inflammatory bowel disease, colorectal cancer, colitis associated cancer, microbiota and immune response

## Abstract

Diseases of the colon are a big health burden in both men and women worldwide ranging from acute infection to cancer. Environmental and genetic factors influence disease onset and outcome in multiple colonic pathologies. The importance of inflammation in the onset, progression and outcome of multiple colonic pathologies is gaining more traction as the evidence from recent research is considered. In this review, we provide an update on the literature to understand how genetics, diet, and the gut microbiota influence the crosstalk between immune and non-immune cells resulting in inflammation observed in multiple colonic pathologies. Specifically, we focus on four colonic diseases two of which have a more established association with inflammation (inflammatory bowel disease and colorectal cancer) while the other two have a less understood relationship with inflammation (diverticular disease and irritable bowel syndrome).

## 1. Introduction 

Inflammatory responses are activated with the help of pattern recognition receptors (PRRs) expressed in both immune and non-immune cells by (i) extrinsic factors such as bacterial or viral structures known as pathogen associated molecular patterns (PAMPs) or (ii) by various endogenous factors that arise due to cellular damage known as danger associated molecular patterns (DAMPs) [1]. Upon stimulation, the crosstalk between cells of the immune system (natural killer (NKs), T and B cells, dendritic cells and macrophages) with non-immune cells activate pro-inflammatory mediators including chemokines and cytokines to generate inflammatory signals. These signals enable the body to recognize, destroy, and clear foreign elements resulting in a successful acute inflammatory response. Inappropriate expression of either pro- or anti-inflammatory mediators can result in immune suppression and initiation of a plethora of chronic inflammatory conditions [2]. This low-grade unresolved chronic inflammation over time is associated with tissue damage and degeneration and is being recognized as a prerequisite for the onset of many neurodegenerative, autoimmune diseases, and cancer [2].

Colonic disease are diseases that affect the colon (large intestine). Pathologies associated with the colon range from colitis, salmonellosis, shigellosis, travellers’ diarrhoea, diverticular disease (DD), inflammatory bowel syndrome (IBS), inflammatory bowel disease (IBD), and colorectal cancer (CRC). DD results in 52–62 cases per 100,000 individuals to be hospitalized in the United Kingdom and the United states, respectively. Moreover, 396 cases per 100,000 individuals are diagnosed for IBD, and approximately 1 million cases of CRC were reported worldwide in 2018. With growth in aging populations, a rise in colonic disease is expected. Pathologies such as salmonellosis, shigellosis, and travellers’ diarrhoea are caused by a bacterial infection and trigger an acute inflammatory process. In contrast, the role of chronic inflammation is becoming more evident in pathologies such as DD, IBS, IBD, and CRCs. Patients with IBD and a subset of patients with acute DD are at an increased risk of developing CRC, mainly due to chronic intestinal inflammation. An increase in the population of patients with IBD and DD, will inevitably increase the risk of cancer development. Medical therapies targeted at reducing mucosal inflammation forms the foundation of treatment in IBD and more recently have been shown to reduce the relative risk of developing CRCs. On the other hand, therapies that result in altering the immune system in the long term can also promote carcinogenesis. Moreover, many patients with IBD and CRC will require treatment for cancer and further treatment for their IBD, which is clinically challenging. The role of the host-immune response and microbiota are recognized as major players in promoting a chronic inflammatory environment within the colon. This review provides an update on the role of inflammation influenced by the host immune response and microbiota in DD, IBD, and CRC.

## 2. Diverticular Disease

Diverticula is a sac-like protrusion that can occur throughout the gastrointestinal tract (including the esophagus [3], the stomach [4], and the small bowel [5]), however, they are most frequently found in the colon [6,7]. DD includes a range of diseases with different manifestations involving the presence of diverticula such as (i) diverticulosis, defined as the presence of asymptomatic diverticula, (ii) symptomatic uncomplicated diverticular disease (SUDD) characterized by persistent abdominal pain, alterations in bowel habits, and absence of obvious diverticular inflammation, (iii) diverticulitis is caused by inflammation of the diverticula. Diverticulitis can be further classified as uncomplicated when diverticular inflammation is localized or complicated when it is associated with other conditions such as fistula, bowel obstruction, perforations, bleeding, and abscesses [6,7]. 

There has been a surge in the incidence of diverticulosis worldwide, with the highest incidence in the United States and Canada, reaching 50% in the populations aged 60 and above [8,9]. Several identified risk factors might play an important role in the increased incidence of diverticulosis. These include genetic predispositions compounded by changes in diet, lifestyle, and prescription drugs (reviewed in detail [6,7,10]). However, the influence of these factors on inflammation and its role in the pathogenesis of DD remains to be understood. In the subsequent sections, we provide an update on how these factors interact and influence the inflammatory landscape observed in DD.

### 2.1. Host Immune Response and Role of Inflammation in Diverticular Disease

Mucosal inflammation triggered by activation of the immune system due to bacterial overgrowth is thought to be the underlying cause of diverticulitis. However, the role of the immune system and mucosal inflammatory mediators in DD is understudied. The underlying pathogenesis of diverticulosis is considered to be a neuromuscular abnormality and increased luminal pressure, with little or no role for inflammation (Figure 1a). Genetic studies have identified a single nucleotide polymorphism (SNP, rs3134646) in collagen type III alpha I chain (*COL3A1*) to be associated with diverticulosis in Caucasian men [11], supporting the role of the altered collagen vascular system (which has an important role in inflammation) observed in diverticulosis. Consistent with the asymptomatic nature of diverticulosis, there was no change observed in the composition of CD4+, CD8+, CD27+, and mass cell tryptase expressing immune cells or in the levels of interleukin-6 (IL-6), IL-10, and tumour necrosis factor-alpha (TNFα) from diverticulosis colonic mucosa biopsies [12]. 

On the other hand, low grade inflammation and visceral hypersensitivity are thought to be an underlying cause of SUDD (Figure 1b). Consistent with this theory, SUDD patients had significantly higher mean lymphocytic cell density compared to patients with diverticulosis [13]. Further characterization of the immune landscape showed an increased number of CD68+ macrophages, but no differences in mast cells or CD3+ T cells in SUDD patients [14,15]. Coherent with the changes in the immune landscape, there was also an increased overexpression of TNFα, IL-6 [16,17], and IL-10 [18] in SUDD patients, with mucosal TNFα levels decreasing during remission [19]. In addition to known inflammatory cytokines, SUDD patients also showed increased expression of tissue inflammatory markers including the basic fibroblast growth factor (b-FGF) and sydecan-1 (SD1) [19], as well as significant upregulation of the neuropeptide neurokinin 1 (NK1) [17], compared to patients with diverticulosis. 

Initial evidence for the importance of the host immune response and role of inflammation in diverticulitis (Figure 1c) came from genetic studies. These include the SNP rs7848647, upstream of the Tumour necrosis factor (TNF)-like cytokine 1A (TL1A)/TNF superfamily member 15 (*TNFSF15*) genes [20]. Other SNPs include: SNP rs4662344 located in the gene for Rho-GTPase-activating protein 15 (*ARHGAP15*), which plays an important role in selective neutrophil functions [21]; the SNP rs7609897 in the gene for collagen-like tail subunit of asymmetric acetylcholinesterase (*COLQ*) important for neuromuscular junctions [21]; a rare SNP encoding a D435N substitution in laminin beta 4 (*LAMB4*) known for its role in the intestinal barrier function [22], and rs67153654 in *FAM155A* (unknown function) [21]. In addition to these SNPs, genome wide association studies (GWAS) from the United States, UK, and Europe identified 25 common susceptibility loci with diverticulitis from genes that were associated with roles in immunity, extracellular matrix biology, cell adhesion, membrane transport, and intestinal motility [23,24]. Additionally, recent evidence from the histopathology analysis of colon from patients with complicated diverticulitis found an increase in the amount of activated CD68+ CD163+ macrophages, which also correlated with steroid intake, compared to those with chronic, recurrent diverticulitis [14]. An enrichment of innate and adaptive immune system pathways at the RNA level were observed in the resected sigmoid tissue of diverticulitis patients compared to non-diverticulosis controls [25]. They further identified four immune-regulatory hub genes: RAS protein activator-like 3 (*RASAL3*), protein tyrosine phosphatase, receptor type C (*PTPRC*), inositol polyphosphate-5-phophatase D (*INPP5D*), and SAM and SH3 domain-containing 3 (*SASH3*) that were associated with diverticulitis [25]. Genetic studies combined with histopathological and gene expression analysis suggest that diverticulitis is associated with deregulation of the immune system.

### 2.2. Influence of the Microbiota on Inflammation Associated with Diverticular Disease 

The influence of the microbiota in progression from diverticulosis to SUDD or diverticulitis (Figure 1) has been hypothesized, however, further investigations are required. Dysbiosis represents a deviation from the healthy gut microbiota homeostasis, resulting in different characteristics and microbiome compositions, influencing disease [1]. With the exception of a lower abundance of *Clostridium cluster IV*, there was a significant overlap in the fecal microbial composition of patients with diverticulosis compared to controls [15]. Another study showed no modification in the fecal microbiota composition of diverticulosis patients compared to controls [26]. Lack of difference between the microbial compositions was also reflected in a study comparing mucosal microbiota from asymptomatic diverticulosis patients compared to controls [27]. In contrast, results from a small study showed a depletion of *Bacteroides fragilis*, *Collinsella aerofaciens,* and *Collinsella stercoris* in fecal samples of patients with DD compared to controls [28]. *Bacteroides fragilis* is known to play a role in the modulation of inflammation [29]. Multiple studies analyzing colonic biopsies from DD patients compared to controls showed a significant enrichment of *Enterobacteriaceae*, a large family of gram-negative bacteria including *Escherichia Coli* (*E.Coli*), *Klebsiella*, *Salmonella*, *Shigella,* and *Yersinia pestis* [25,30]. Discrepancies in these studies can be attributed to small sample sizes, differences in detection methods, and disease burden being studied. Despite these limitations, results from these studies suggest that diverticular pockets may promote the selection of specific microbial communities that modulate the tissue environment and promote progression of diverticulosis towards more symptomatic diseases such as SUDD and diverticulitis. 

Stool samples analyzed from SUDD patients are associated with selective depletion and overgrowth of taxa influencing the severity of inflammation. This is reflected from findings across different studies conducted on fecal samples from SUDD patients. Examples include a higher abundance of *Ruminococcus* and lower abundance of *Roseburia* associated with greater bloating severity [31], an abundance of *Cyanobacterium* positively correlated with pain intensity [31], and a lower abundance of *Clostridium cluster IV*, *Fusobacterium,* and *Lactobacillaceae* compared to controls [15]. *Lactobacillaceae* is associated with anti-inflammatory and immune regulatory effects in colitis models [32], thus its selective loss can influence intestinal inflammation in SUDD patients. Results from fecal samples of four SUDD patients showed an attenuation of *Roseburia*, *Veillonella*, *Haemophilus,* and *Streptococcus* after a rifaximin antibiotic treatment [33]. Another study conducted using fecal samples from SUDD patients reported an increased abundance of *Akkermansia* [26], and is associated with a mucin stimulated metabolic substrate as a result of colonic inflammation [34]. Contrary to the findings from fecal samples, there was a significant reduction of *Akkermansia* observed from the colonic mucosal biopsies of SUDD patients [15]. Despite limited evidence, results from the above studies suggest an interplay between mucosa-associated microbial communities and the immune system in regulating the severity of intestinal inflammation in SUDD patients. 

Alterations in the gut microbiome has been speculated to at least partially contribute towards inflammation and diverticulitis progression. An overgrowth of the *Pseudobutyrivibrio*, *Bifidobacterium,* and *Christensenellaceae* family was observed from fecal samples of an individual with history of recurring acute diverticulitis [31]. In this study, they also found that the overall microbiome diversity positively correlated with a marker of intestinal mucosal inflammation, fecal calprotectin [31]. Overrepresentation of *Bifidobacteria*, namely *Bifidobacterium longum* was also observed in the colonic mucosal samples of acute diverticulitis patients [35]. Findings from a cross-sectional study analyzing the microbial composition of fecal samples from diverticulitis patients showed that there was a significant overgrowth of *Enterobacteriacae* [36]. The studies provide preliminary evidence of a deep homeostatic disruption of mucosa-associated microbial communities in diverticulitis. 

Notwithstanding the limitations, the above studies provide prima facie evidence for the involvement of microbiota in the progression but not in the pathogenesis of diverticulosis to SUDD or diverticulitis. This is supported by the fact that the microbial composition did not differ between controls and patients with asymptomatic diverticulosis, while significant changes in the microbiome were observed in SUDD and diverticulitis patients. These changes are reflective of changes in the gut microenvironment and are capable of modulating inflammation; however, whether they contribute directly or are merely bystanders to the pathophysiology of DD remains to be determined.

## 3. Inflammatory Bowel Disease

IBD is a chronic inflammatory disease of the gastrointestinal tract [37]. IBD is classified into two subtypes: ulcerative colitis (UC) and Crohn’s disease (CD). UC affects the colon and CD may affect any region of the gastrointestinal tract, but occurs primarily in the terminal ileum of the small intestine. IBD has become a global disease with accelerating incidence in newly industrialized countries [38]. The age-standardized prevalence rate of IBD has risen globally from 79.5 per 100,000 population in 1990 to 84.3 per 100,000 population in 2017 [38,39]. Several large longitudinal studies have shed some light on the environmental risk factors of IBD. A consistent message from these studies is that there appears to be a lower risk of IBD in people who consume more fruits and vegetables, and increased risk is associated with consuming less fruits and vegetable and eating more animal fat [40,41,42,43]. A high intake of fiber from fruits and vegetables was associated with a reduced risk of CD [40]. A high intake of omega-6 fatty acids and low consumption of marine omega-3 was associated with an increased risk of IBD [42,44]. Prebiotics such as inulin-type β-fructans are non-digestible carbohydrates that beneficially alter the activity of gut microbiota [45]. In a recent pilot study, it was reported that a high dose of fructans resulted in functional but not compositional shifts of the gut microbiota associated with a reduced colitis score in the high dose UC group when compared to controls [46]. IBD is a complex disease involving genetic susceptibility, intestinal microbiota dysbiosis, the host immune system deregulation, and environmental factors such as diet and stress, all of which contribute to a cycle of chronic inflammation in the intestine [37]. 

### 3.1. Host Immune Response and Role of Inflammation in IBD

Antigens from food and bacteria come into contact with dendritic cells and macrophages in the lamina propria of the intestine which process the antigens and induce oral tolerance [47]. Macrophages in the lamina propria are increased in IBD patients particularly in active lesions [47]. The increase appears to be linked with an increase in the frequency of immature macrophages [48]. This seems to be due to the inflammatory intestinal environment in IBD which increases the recruitment of monocytes [49]. These monocytes are maintained in the immature pro-inflammatory state which amplifies the intestinal chronic inflammation [47]. A recent GWAS study identified genes involved in macrophage maturation, as well differences in the gene expression of genes and promoters linked to monocyte activation and differentiation to be enriched in IBD [50]. The study further supports the importance of monocyte maturation in IBD risk.

Several inflammatory pathways and cytokines have been implicated in the pathogenesis of both CD and UC. One of the most studied inflammatory pathways in IBD is the nuclear factor kappa-light-chain-enhancer of activated B cells (NF-κB) signaling pathway which involves hyper-activation in epithelial or immune cells from IBD patients [51]. A recent study has shown that macrophages isolated from IBD patients were segregated into hypo-responsive NF-κB for UC patients and mostly hyper-responsive NF-κB in CD patients [51]. A highly upregulated cytokine in IBD is interferon-γ (IFNγ). The role of IFNγ in IBD has been mainly attributed to immune modulation [52]. More recently, it was demonstrated that IFNγ exerts potent activity on vasculature. IFNγ was shown to be expressed and activated in IBD inflamed mucosal biopsies and this seemed to be acting on blood vessels in endothelial cells resulting in an increased blood vessel density [53]. More recently, a study has shown that inhibition of IFNγ results in disruption of the vascular barrier and inhibition of IFNγ ameliorates experimentally induced colitis in IFNγ knockout mice [54]. A TNF-like ligand 1A (TL1A) has been shown to be an important mediator of intestinal inflammation [55] as well as mucosal healing through an ILC3-IL-22 axis [56]. The levels of TL1A are increased in IBD patients [55]. In one study, it was shown that macrophages from CD patients produced a higher level of TL1A, resulting in an increased production of IFNγ from T cells [57]. More recently, it has been demonstrated that polymorphisms in the gene (*TNFSF15*) were associated with an increased risk in IBD and that reducing the expression of this gene on monocytes and macrophage is associated with increased susceptibility [58]. Overall, these studies further support the complex interplay of multiple inflammatory and immune pathways in IBD (Figure 2). 

### 3.2. Influence of the Microbiota in Inflammation Associated with IBD

The pathogenesis of IBD revolves around interactions between host immunity and the gut microbiota and genetic variants in IBD susceptibility genes. Recent work has aimed at understanding these interactions better. IBD is known to have a strong genetic component based on genetic analyses and twin studies [59]. It is estimated that the heritability of IBD on the liability scale is 0.75 for CD and 0.67 for UC [60]. There are over 240 IBD risk genetic loci identified and many of these loci appear to be involved in the response to microorganisms [61,62,63]. These include SNPs in *NOD2 (Nucleotide-binding oligomerization domain 2)*, *CARD9 (Caspase Recruitment Domain Family Member 9)*, *ATG16L1 (Autophagy related 16 like 1)*, *IRGM (Immunity Related GTPase M),* and *TNFSF15* [61,62,63]. Studies have been performed looking at the interplay between genetic variation and the gut microbiota [64]. In one study, an association between the *NOD2* (a gene important in innate immunity) risk allele and the relative abundance of *Enterobacteriaceae* was found [65]. In another study, CD patients with the IBD risk allele *ATG16L1-T300A* in the *ATG16L1* gene (important in innate immunity and inflammation) was associated with impaired clearance of patho-symbionts in ileal inflammation [66]. More recently, a study using the linear regression model and replication analysis showed a negative association between *NOD2* risk alleles and the *Roseburia* genus and the *Faecalibacterium prausnitzii* species in the gut microbiota [64]. The authors further explored causal models between genetic variation-bacteria-IBD and found microbiota manipulation as a potential mechanism underlying the association between IBD and the genetic variants in *NOD2* and *ATG16L1* [64].

The gut microbiota is known to have a role in modulating the immune system. The production of IL-10, which downregulates IFNγ responses, requires the microbiota [67]. The microbiota is also important for the activation of intestinal regulatory T (Treg) cells which are absent in germ free mice [68]. In one study, the microbiota was required for proper intestinal barrier repair through innate lymphoid cell production of IL-22 [69]. More recently, it was demonstrated in mice that an intact microbiota is required for the reduction in inflammatory T helper 1 (T_h_1) cell responses against food and the microbiota itself, and promoted antigen presenting cell driven Treg differentiation [70]. On the other hand, microbiota disruption promoted a specific T_h_1 cell expansion and contributed to tissue pathology and infection [70]. Treg cells induced by the microbiota contributes to gut homeostasis. Most of the work has been done in mice [71]. However, Treg cells are abundant in healthy human colons and decreased in IBD. A recent study showed that human *Clostridium Faecalibacterium prausnitzii (F.prausnitzii)* which is decreased in CD, influences dendritic cells to secrete Treg polarizing molecules and inhibited their pro-inflammatory production [71]. Increased interleukin-13 (IL-13), an inflammatory cytokine produced by immune cells, has been previously implicated in gut inflammation and ulcerative colitis [72]. More recently, it has been shown that a subset of UC patients that have been characterized with elevated 1L-13 mRNA, which was associated with more extensive colitis, had a prevalence of mucosa-associated microbiota genera *Prevotella* and lower *Sutterella* and *Acidaminococcus* compared to patients with a low IL-13 mRNA content [73].

Interestingly, the relationship with *Sutterella* species and UC in relation to inflammation and immunity is a complex one. In one study, fecal microbiota transplantation (FMT) showed an association between UC remission and enrichment of the species [74,75]. However, other studies show an inverse relationship between the presence of *Sutterella* and inflammation [73,76]. These contrasting observations were later explained based on *Sutterella* being a commensal bacteria with an immunoglobulin A (IgA) degrading ability and thus rather than directly inducing inflammation, the bacterial species could impair the antibacterial immune response [77].

Overall these results show a complex and intricate relationship between the gut microbiota, inflammation, and immunity in relation to IBD (Figure 2), and highlight that while the broad microbiota analysis gives a general idea of the involvement of bacteria, a more specific analysis of the interplay between bacterial groups and host immunity is required for a more effective treatment of the disease.

### 3.3. Irriteable Bowel Syndrome Link to IBD

IBS is a disorder where affected individuals exhibit abdominal discomfort and irregular bowel movement with either predominantly diarrhea, constipation, or both [78]. Since there are no clear explanations for the symptoms exhibited, IBS is classified as a functional gastrointestinal disorder (FGID) [79,80]. A study in 2010 has built on observations that gastrointestinal symptoms in IBD during remission reflect a coexistence of IBS [81]. The study found that IBS-type symptoms were common in both CD and UC patients who are considered to be in remission. When they measured the fecal calprotectin (a marker of intestinal inflammation), they observed that it was highest in patients with IBS-type symptoms compared with healthy controls and those patients without IBS-type symptoms. The authors suggest that this might be a result of an ongoing underlying IBD activity.

The role of inflammation in IBS is still unclear; however, over the years ample evidence has accumulated in relation to some of the mechanisms associated [82]. Genetic links to IBS has indicated a role of inflammation in the disease. A recent meta-analysis and systemic review has confirmed two polymorphisms (SNPs rs4263839 and rs6478108) in the *TNFSF15* gene for a proinflammatory cytokine were associated with an increased IBS risk. On the other hand, SNP rs1800896 in IL-10 an anti-inflammatory cytokine, was associated with a decreased risk of IBS [83]. Low-grade inflammation has been reported in IBS, which can stimulate visceral nerves and induce dysmotility [84]. More recently, it was reported that IBS patients with diarrhea had increased prostaglandin E2 gene and protein expression in mucosal biopsies. Another study showed higher serum IL-6, IL-8, and TNF-α in IBS, suggesting a role of systemic inflammation [85,86].

Since the 1960s, gastrointestinal infections have been reported to be risk factors for an IBS subtype known as post-infectious IBS. More recently, a systematic review and meta-analysis of different gastrointestinal outbreaks found the incidence of post-infectious IBS ranging from 3.7 to 36% and lasting up to eight years [87]. Post-infectious IBS shows similarities to IBS with diarrhea however, unlike sporadic IBS, inflammation and mucosal injury have a more significant role [88]. More recently, a 3-fold increase in the risk of non-constipating IBS was reported in individuals with the presence of colonic spirochetosis [89]. The presence of colonic spirochetosis was characterized by an increase in eosinophil and lymphoid follicles determined by colonic biopsies [89]. Increased macrophages, T lymphocytes, and serum IL-6 are hallmarks of post-infectious IBS [88]. Additionally, post-infectious IBS patients have dysbiosis but a different profile to those with IBS [90]. 

Recent studies have demonstrated evidence of low-grade mucosal inflammation and a link to the gut microbiota in some IBS patients similar to IBD. The use of antibiotics provided a connection for the role of the gut microbiota in IBS. Previously a case-control study has reported that antibiotic use was linked to an increased risk of developing IBS [91]. More recently, 83% of patients who developed GI symptoms reported antibiotic use [92]. Changes in the microbiome and the development of IBS has been reported extensively (reviewed in Pimentel et al. [93]). Numerous studies have compared the intestinal microbiota in IBS to a healthy control and found a lower microbial diversity and an increase in firmicutes-to-bacteroidetes ratio in IBS subjects [93]. Data suggest that an increased methane production as a result of archeal microbial species such as *Methanobrevibacter smithii* can influence intestinal motor activity leading to constipation. A placebo-controlled trial showed that an antibiotic combination used to eradicate methane from the breath resulted in significant improvements in symptoms of constipation in IBS [94]. A recent consensus now considers methane in the breath as an accurate indicator of intestinal colonization with methane producing microbes and is an important tool for assessment of constipation in IBS [95]. 

It is unclear whether IBS is part of the IBD spectrum or a separate disease [96]. This lack of clarity may make it difficult to differentiate between IBS-like symptoms and an upcoming flare in IBD patients, complicating the clinical management of patients with IBS [97]. Results from a recent study, inflammatory proteins from the serum of UC patients with and without IBS symptoms were found to be different from those of IBS patients and healthy controls. The proteins that were identified as most important for differentiating the groups were caspase-8 (CASP8), axin 1 (AXIN1), sulfotransferase 1A1 (ST1A1), and tumour necrosis factor superfamily member 14 (TNFSF14), which all were found in higher levels in UC as compared with non-UC subjects [97]. Results from this study warrants further investigation into the role of inflammation and microbiota to distinguish IBS from IBD.

## 4. Colorectal Cancer

CRC is one of the major malignancies in humans and is the third most common cause of cancer-related deaths affecting one million people worldwide [98]. About 5–10% of CRCs are inherited and include hereditary non-polyposis colorectal cancer (HNPCC, lynch syndrome) and familial adenomatous polyposis (FAP) [99]. Another 5% of CRCs can develop from patients that have IBD and is known as colitis-associated cancer (CAC) [99]. However, a large proportion of CRCs develop sporadically and can be classified into six sub-groups based on the presence of microsatellite repeats, chromosomal instability, and methylation [100]. Several factors including genetics, diet, and lifestyle contribute towards the risk of developing CRC (reviewed in detail [99]). The influence of these risk factors in modulating inflammation during the pathogenesis of the disease remains unknown, however, anti-inflammatory medication either prevents or delays the development of CRC in hereditary and sporadic cases [101]. Thus, it is becoming evident that chronic inflammation plays an important role in both the development and progression of CRC.

### 4.1. Host Immune Response and Inflammation in Colon Cancer

Inflammation is an established hallmark of cancer [102], however, it is a double-edged sword. Cross-talk between cells of the innate and adaptive immune system along with intestinal epithelial cells and stromal cells through a network of cytokines, chemokines, and growth factors contribute towards cancer-associated inflammation. Anti-tumour responses can be obtained by targeting tumour cells with cytotoxic T-lymphocytes (CTLs) or reducing non-specific inflammation by T-regs [103]. On the other hand, elevation in non-specific inflammation caused by a barrier dysfunction of the intestinal epithelial cells and their pathogenic interaction with microbiota and the immune system can promote tumourigenesis [104,105,106]. 

Disruption of the barrier function of the gut intestinal epithelial cells causes the loss of the protective shield between the host and hazardous intestinal lumen. Damage to the barrier leads to the loss of homeostasis, pathological interaction between epithelial cells, microbiota, and the immune system resulting in tumourigenesis over time [104,105,106]. A barrier dysfunction causes activation of type I interferons and myeloid differentiation primary response 88 (MyD88) by exogenous pathogens, microbial metabolites, and DAMPs. A pathological interaction between epithelial cells and the immune system can result in accumulation of DNA damage and accumulation of nucleic acid fragments [107], which in turn can activate the innate immune system. One such example is activation of type I interferon response via activation of the stimulator of interferon genes (STING) pathway in Ataxia-Telangiectasia Mutated (ATM) deficient mice that are incapable of activating DNA repair pathways [108,109]. Another example comes from STING-deficient mice that resulted in an increased NF-kB activation and had a higher tumour load when treated with the carcinogen, Azoxymethane (AOM) followed by induction of intestinal inflammation by dextran sulfate sodium (DSS) [108,109]. It was recently found that the intestinal vascular endothelium acts as a second barrier in the gut against the outside environment [110]. During IBD, inflammation has been shown to activate angiogenesis [111,112]. In line with this, increased levels of angiogenic growth factors, including vascular endothelial growth factor A (VEGF-A), placental growth factor (PGF), and platelet-derived growth factor (PDGF), have been detected in the inflamed mucosa and the blood of patients with active disease [111,112]. IFNγ was shown to activate Guanylate binding protein-1 (GBP-1), a key mediator of angiostatic effects of inflammation in both IBD [53] and CRC [113]. Results from human studies and mice models highlight the importance of chronic inflammation leading to CAC.

Pathological interaction between microbial products and tumour-associated myeloid cells can increase the production of IL-23, which in turn can induce T_h_17 cell activation in mouse models of CAC and sporadic CRCs [114]. Polarization of T_h_17 and related cytokines including 1L-17A, IL-17F, IL-21, IL-22, TNF-α, and IL-6, in CRCs is associated with tumour promoting inflammation, immunosuppression, induction of pro-angiogenic factors, and poor patient outcome [115,116]. The importance of T_h_17 signaling comes from multiple studies. Knockout of *Il17ra*, a key effector of T_h_17 response, resulted in reduced tumour burden in a CAC mouse model [117]. In another study using the CAC mouse model, depletion of innate lymphoid cells, IL-22, IL-17, or IFN-γ resulted in decreased tumour burden [118]. Elevation of IL-6 is also known to induce T_h_17 cell differentiation while activating signal transducer and activator of transcription 3 (*STAT3)* signaling in epithelial cells to promote tumourigenesis in a CAC mouse model [119,120]. Another member of the IL-6 family, IL-11 is also associated with *STAT3* activation in human samples [121]. Elevated levels of IL-22 produced by CD4+ T cells also induced epithelial *STAT3* activation and promoted cancer stemness. Finally, inhibiting STAT3 signaling with NT157, reduced tumour burden in a sporadic mouse model of CRC [120]. Collectively, these results suggest that activation of STAT3 plays an important role in immune evasion and progression of CRC.

With the exception of CAC, it is becoming evident that chronic inflammation follows tumourigenesis in CRCs. The presence of activated inflammatory cells in the tumour microenvironment can produce reactive oxygen species (ROS) and reactive nitrogen intermediates that can induce accumulation of additional mutations and epigenetic changes [107]. A loss of IL-10 from the T-cell population resulted in impaired expansion of T_h_1 cells and failed to mount an IFN-γ dependent cytotoxic response resulting in polyposis and invasive carcinomas in an Adenomatous polyposis coli (Apc) Δ468 mutation mouse model of sporadic CRCs [122,123]. Apc^MIN^ mice treated with IL-10 resulted in decreased polyp burden and disease severity [103]. Taken together, these studies suggest a crucial role of IL-10 in tumour surveillance. The importance of IFN-γ dependent cytotoxic response by T_h_1, CTLs and NK cells [124] towards tumour cells is further established in Apc^MIN/+^ mice, where deletion of one copy of IFN-γ resulted in faster progression of colonic adenocarcinomas [125].

Another important player in the regulation of inflammation is NF-kB signaling. A loss of NF-kB from the epithelial cells resulted in reduced tumourigenesis [126]. In contrast, prolonged activation of NF-kB in the epithelial compartment resulted in accelerated tumour development of CRCs resulting from increased DNA-damage and upregulation iNOS [127]. A loss of NF-kB signaling in the myeloid compartment also slowed tumour growth due to the reduction in pro-inflammatory mediators. Interestingly, NF-kB promotes tumourigenesis when expressed in intestinal mesenchymal cells of a CAC model but not a sporadic CRC [128]. Altogether, these results suggest that NF-kB plays an essential role in modulating inflammation in CRCs. Thus, taken together this section highlights the role of cytokines, chemokines, and survival factors that can promote tumour growth by suppressing immune-mediated tumour elimination (Figure 3). 

### 4.2. Influence of Microbiota in Inflammation Associated with Colon Cancer

The role of intestinal microbiota in regulating inflammation and development of CRC has become evident. Bacteria can contribute towards inflammation during the development of CRC in four ways. These include loss of barrier function and consequently induction of tumour-promoting inflammation; alteration in the composition of gut microbiota or its metabolites causing tumour-promoting inflammation; the presence of genotoxic bacteria can promote colonic inflammation and cancer by inducing DNA damage and epithelial cell transformation. 

The main features of CRCs in mice and human are a disrupted barrier function characterized by the loss of production and localization of tight junctions and no mucus production [114]. These features result in invasion of bacteria and bacterial degradation products into the tumour stroma of both mice and humans [114]. Commensal and pathogenic bacteria (*Citrobacter rodentium* and *Bacteroides fragilis*) and their products can engage with Toll-like receptors (TLRs) on tumour infiltrating myeloid cells and activate MyD88 mediated production of IL-23 and tumour promoting inflammatory cytokines such as IL-17A, IL-6, and IL-22 [114]. Additionally, mice defective in mucin glycosylation develop CRCs in the presence of gut bacteria [129]. Mice lacking N-formylpeptide receptors on the surface of the epithelial cells are associated with an increased CAC development [130]. Moreover, ablation of bacterial sensing TLR2, 4, 9, and MyD88 in hematopoietic cells resulted in a decreased CRC development in mice [114,131,132]. Taken together these studies demonstrate that barrier defects promote CRC development and progression by inducing tumour promoting inflammation.

An imbalanced state of commensal bacteria or dysbiosis is commonly encountered in patients with inflammatory disease and CRC [133]. Altered commensal bacterial populations have been observed in Apc^MIN^ mice even before polyp visibility, suggesting an importance of host-bacterial interaction in colon tumorigenesis [134]. Another example of bacterial dysbiosis comes from the altered microbial content observed in the feces of AIM2-deficinet mice [135]. Studies conducted in NOD2 [136,137], IRAK-M, and IL-33 [138] deficient mice and mice lacking the anti-microbial peptide adrenomedullin [139], have an altered microbiota composition and increase CAC susceptibility. Further evidence comes from studies using probiotics to alter the course of CRC development. Probiotics treatment with a cocktail of *Lactobacillus acidophilus*, *Bifidobacteria bifidum*, and *Bifidobacteria infantum* reduced CRC in rats and mice [140,141]. Moreover, a high intake of red meat has been demonstrated to alter the microbiota composition, which in turn may contribute towards CRC development [142]. Both diet and genetic caused mutations also altered the microbial composition and are associated as risk factors for CRC development [143]. Results from these studies provide evidence that the altered microbial composition can influence the course of CRC development.

An advent of next generation sequencing techniques have enabled the identification of virulence-associated bacterial genes in the tumour microenvironment of CRCs, suggesting a role for these bacteria in driving CRC [144]. An *Helicobacter hepaticus* infection in multiple mouse strains that harbor defects in immune components result in the development of CRC [145]. Examples include Rag2 knockout mice infected with *Helicobacter hepaticus* that develop severe gut inflammation and cancer due to an increase in nitric oxide production caused by accumulation of macrophages and neutrophils in their colons [146,147]. Another study has demonstrated that an *Helicobacter hepaticus* infection results in upregulation of expression of pro-inflammatory cytokines including IL-1α, IL-1β, IL-6, IFN-γ, and TNF-α and development of CRC of SMAD family member 3 (SMAD3) knockout mice [148]. Apc^MIN^ mice infected with human *Enterotoxigenic Bacteroides fragilis* increased IL-17A production from T_h_17 cells and γδ T cells [149] and accelerated CRC development [150]. Analyses of human CRCs have found frequent association of the *B. fragilis* toxin, a 21 kD virulence factor, with poor prognosis [151,152,153]. Another bacteria found enriched in CRCs compared to a matched normal tissue is *Fusobacterium nucleatum* and its presence in CRCs was also associated with poor prognosis [154]. Once again, human CRCs enriched for a *Fusobacterium nucleatum* infection associated with elevated levels of IL-17A along with TNF-α [155]. The Apc^MIN^ mouse model infected with *Fusobacterium nucleatum* demonstrated that this infection promotes CRC development by acting on both tumour cells, enabling expansion of immune cells, and preventing T cell mediated immune killing of CRC cells [154,155]. Other bacteria implicated in the CRC development include *Peptostreptococcus anaerobius*, sulfidogenic bacteria [154,155], and *Salmonella* [156]. These results suggest that pathogenic bacterial infection can trigger a process of chronic inflammation resulting in accumulation of DNA damage because of repeated cycles of tissue damage and regeneration eventually resulting in tumour development.

The presence of genotoxic bacteria can induce DNA damage and promote malignant transformation of the intestinal epithelial cells. *Enterococcus faecalis* is a commensal bacteria that has the potential to produce extracellular superoxide. In vitro co-culture experiments have demonstrated the ability of *Enterococcus faecalis* to promote chromosomal instability in epithelial cells [157]. Moreover, macrophages infected with *Enterococcus faecalis* induce production of a superoxide capable of inducing DNA damage in the epithelial cell and upregulation of cyclooxygenase (COX)-2 in the macrophages, both of which promote accumulation of mutations and malignant transformation [157]. A potentially harmful member of the gut bacteria, polyketide synthetase (pks)+ *E Coli*, produces a genotoxin called colibactin, which has been associated with the onset of CRC [158]. Recently, a unique mutation signature caused by colibactin-producing *E.Coli* was identified in human CRC tumours and metastases [159]. Another strain of *E.Coli* carrying the cyclomodulin virulence factor also enhanced the tumour load in Apc-min mice [160]. These data suggest that the enrichment of genotoxic commensal bacteria predispose the gut for tumour development caused either by or as a consequence of chronic inflammation.

## 5. Conclusions

Understanding the role of specific inflammatory mediators in each of these colonic diseases is essential for improving screening, diagnosis, and therapeutic strategies for patients. Even though there are essential and common inflammatory cytokines including IL-6, IL-10, TNF-α, and pathways including NF-kB are recognized to be drivers of inflammation across all four colonic diseases, the relative contribution, crosstalk, and cell-type specific effects are only starting to be comprehensively defined. 

Additionally, there is a growing evidence for the role of the gut microbiota and its contribution towards the pathogenesis of inflammation in all four colonic pathologies. Thus, there is an urgent need to understand the complex roles of inflammatory signaling cascades and the cell types involved in driving the underlying pathology. This is now possible with the advent of CRISPR/Cas9 approaches, 3D models, and patient derived cultures. These techniques will enable us to improve our understanding of the intimate relationship between intestinal epithelial, gut microbiome, and inflammatory cells in order to personalize care and improve therapeutic intervention for patients with underlying colonic pathologies.

## Figures and Tables

**Figure 1 ijms-21-04748-f001:**
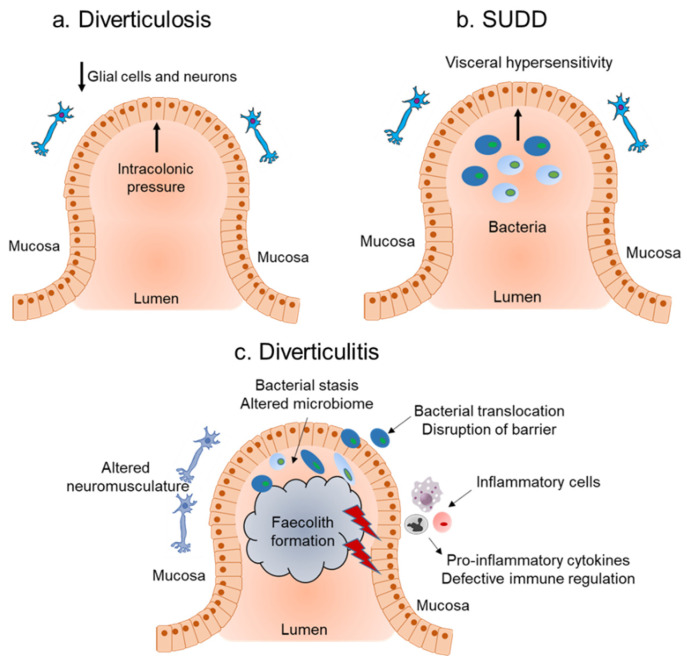
Pathophysiology of diverticular diseases. (**a**). Diverticulosis is postulated to be caused by an increase in the intracolonic pressure from the lumen compounded by neuromuscular abnormalities. (**b**). SUDD (symptomatic uncomplicated diverticular disease) can be caused by alterations in microbiota, the presence of low-grade inflammation, and increased visceral hypersensitivity. (**c**). Diverticulitis is proposed to be caused by alterations in the intestinal microbiota leading to disruption of the mucosal barrier, translocation of the bacteria leading to inflammation, and/or due to local trauma from a faecalith.

**Figure 2 ijms-21-04748-f002:**
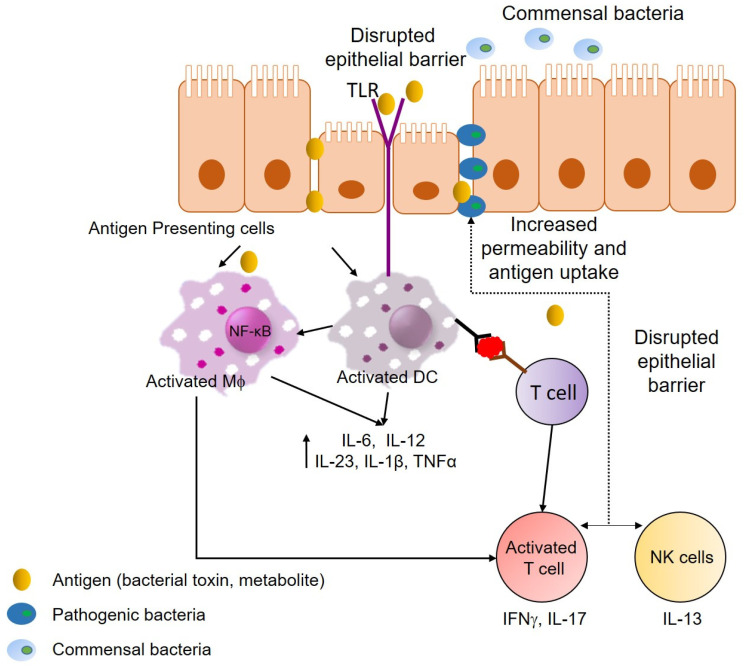
Pathophysiology of inflammatory bowel disease. Disruption of the epithelial layer causes increased permeability of luminal antigens through the epithelial layer. These antigen presenting cells (macrophages—Mϕ and dendritic cells—DC) are activated by recognition of antigens via toll-like receptors (TLR). Activation of the antigen presenting cells results in activation of the nuclear factor kappa-light-chain-enhancer of activated B cells (NF-κB) pathway, which stimulates the production of inflammatory cytokines (interleukin (IL)-6, IL-12, IL-23, IL-1β, and TNFα) via increased transcription. Antigen presenting cells then process the antigen and present it to T cells, in turn promoting their differentiation and leading to production cytokines (IFNγ and IL-17) that are responsible for moderating cell-mediated immunity. Natural killer cells (NK cells) participate in surveillance to infectious pathogens and are the main source of IL-13, which is responsible for disruption of the epithelial barrier.

**Figure 3 ijms-21-04748-f003:**
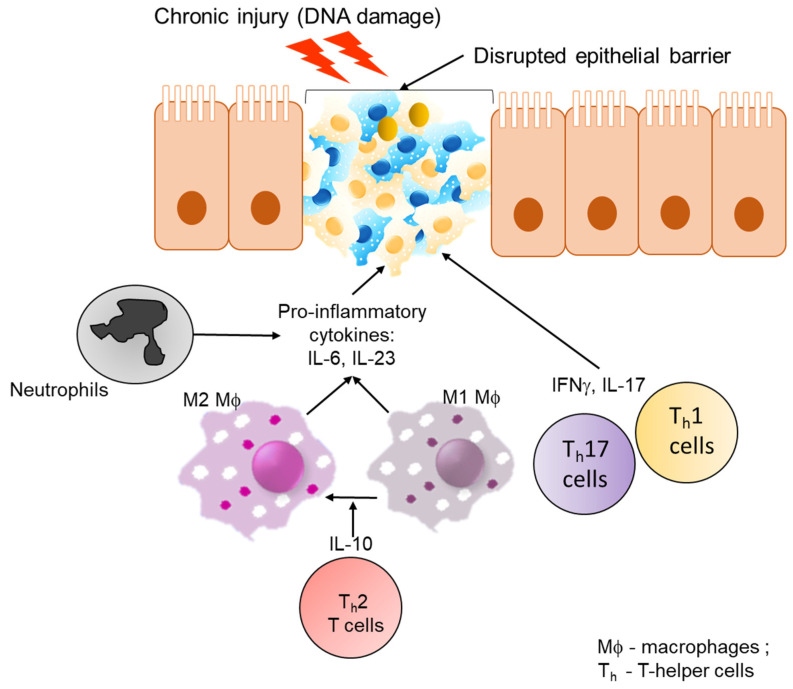
Role of inflammation and microbiome in colorectal cancer. Chronic DNA-damage caused either by infection, prolonged inflammation, and tissue injury due to barrier disruption results in activation of NF-κB signaling in epithelial cells promoting sustained cell proliferation and evasion of apoptosis. This eventually results in oncogenic transformation of epithelial cells. Sustained signaling of IL-10 from T_h_2 T cells results in differentiation of macrophages from M1 (drives classical inflammatory response) to M2 (promotes immune suppression and wound healing). M2 macrophages and neutrophils secrete pro-inflammatory cytokines IL-6 and IL-23. IL-23 promotes accumulation of T_h_17 T cells, which secrete IL-17 which is known to play an important role in colitis associated cancer. IL-6 levels directly contribute to the development of colorectal cancer by activation of NF-κB and STAT3. T_h_1 T cells produce IFNγ, as a protective mechanism to deal with ongoing tissue damage caused by the presence of the tumour. Mϕ—macrophages; T_h_—T-helper cells.

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
