# Peer review of "Role of Inflammation in Pathophysiology of Colonic Disease: An Update"

_ijms, 2020, doi:10.3390/ijms21134748_

Round 1
Reviewer 1 Report
This review described the role of inflammation in a selected set of colonic diseases. The relevance is high as both host and microbial factors are discussed. The reviewer has major and minor comments.
Major comments:
Keywords: missing.
Methods: not described. Is this a narrative review or was a literature search performed based on prespecified search terms?
Diverticulosis: what about the potential role of spirochetes? http://dx.doi.org/10.1016/j.humpath.2014.10.026
IBS, 3.1 line 217: what is the role of bile for tryptase?
IBS, 3.3: most recent evidence clearly points out that IBS and IBD are different, in contrast to 2010 evidence and this should be mentioned more clearly.
Most sections are rather long and also (partly) overlapping, so please revise and shorten eg. IBS-IBD and CRC-CAC (with overlapping immune pathways).
Minor comments:
Please note the double/multiple spaces:
- Line 28
- Line 68
- Line 183, also remove “that”
- Line 256, no capital A
- Line 324: host immunity,
Please note typo’s/mistakes:
- Line 225: pathogenies
- Line 251: alteration not alternation?
- Line 252: reference 46 bold and italic?
- Line 352: homoeostasis
- Line 354 human not italic
- Line 357: IL-3 instead of 13?
- Line 362: strutella?
- Line 428: burdgen
- Line 431: “can in the tumour”
- Line 446: tumourogenesis or tumorigenesis (line 473)?
- Line 461: “humanS”
- Line 472: host-bacterail
- Line 510: “induce production”
Also please rephrase the following:
- Line 137-138 (link between SUDD and “metabolic substate” unclear)
- Line 152 “deep disruption homeostasis”
- Line 153 “studies, provide…”
- Line 156-157 “While..” at beginning sentence
- Line 161 “where individuals affected,”
- Line 175: “include,”
- Line 227 “provide a glimpse into” could be replaced with “suggested”
- Line 334 “it was identified that”
- Line 341-342 “microbiota mediation..”?
- Line 511 “both of..”?
- Line 521 “While..” at beginning sentence
Other comments:
- Line 10: “diseaseS of the colon ARE” or plural
- Line 199: define abbreviation at first mention
- Line 231: first author is Pimentel, not Lembo
- Line 231: “intestinal microbiota IN IBS to”
Author Response
We would like to thank the reviewer for their insightful comments and for thoroughly going through our manuscript. We have addressed all of the concerns raised by the reviewer. Please find point by point responses detailed below:
Keywords: missing: We have now added the keywords to the revised version of the manuscript.
Methods: not described. Is this a narrative review or was a literature search performed based on prespecified search terms? We have not described a method section as this a narrative review.
Diverticulosis: what about the potential role of spirochetes? http://dx.doi.org/10.1016/j.humpath.2014.10.026: The paper suggested by the reviewer shows the potential role of colonic spirochetes in IBS and this has been included in the IBS section ( 3.3 Lines, 300-302).
IBS, 3.1 line 217: what is the role of bile for tryptase? Tryptase from mast cells is not associated with bile. We apologize for the confusion as a result of a missing reference [Ref 63 in the revised version]. In line with advice from reviewers 1 and 2 we have now shortened the section on IBS and combined it with IBD. In ordere to do so, the IBS section has now been modified and this paragraph removed.
IBS, 3.3: most recent evidence clearly points out that IBS and IBD are different, in contrast to 2010 evidence and this should be mentioned more clearly. We have included recent study from 2020, discriminating IBS from IBD and expanded on these points in section 3.3 (lines 319 - 327)
Most sections are rather long and also (partly) overlapping, so please revise and shorten eg. IBS-IBD and CRC-CAC (with overlapping immune pathways). In line with reviewer 1 and 2’s comments, we have removed the IBS section and added as an IBD subsection (section 3.3). CAC is already part of the CRC section.
Minor comments:
Please note the double/multiple spaces:
· Line 28 - Done
· Line 68 - Done
· Line 183, also remove “that” IBS section modified: Section has been modified
· Line 256, no capital A: IBS section modified
· Line 324: host immunity,: added “and”
Please note typo’s/mistakes:
· Line 225: pathogenies section removed
· Line 251: alteration not alternation? Section removed
· Line 252: reference 46 bold and italic? Section removed
· Line 352: homoeostasis Corrected spelling
· Line 354 human not italic Done
· Line 357: IL-3 instead of 13? Changed
· Line 362: strutella? Changed too Sutterella
· Line 428: burdgen: Changed to burden
· Line 431: “can in the tumour”: Changed to “in the tumour”
· Line 446: tumourogenesis or tumorigenesis (line 473)?: Has been changed to “tumorigenesis”
· Line 461: “humanS”: Corrected
· Line 472: host-bacterail: Changed to host-bacterial
· Line 510: “induce production”: Corrected to “induce production”
Also please rephrase the following:
· Line 137-138 (link between SUDD and “metabolic substate” unclear): Changed to “mucin associated metabolic substrate”
· Line 152 “deep disruption homeostasis”: Changed to “ deep homeostatic disruption..”
· Line 153 “studies, provide…”: Changed to “ above studies provide”
· Line 156-157 “While..” at beginning sentence: Changed.
· Line 161 “where individuals affected,” changed to “where affected individuals”
· Line 175: “include,” section removed
· Line 227 “provide a glimpse into”: Changed to “ provided a link for the role of the gut..”
· Line 334 “it was identified that” :Changed to “found that”
· Line 341-342 “microbiota mediation..”? Changed to “microbiota manipulation..”
· Line 511 “both of..”?: Changed to “both of which..”
· Line 521 “While..” at beginning sentence Changed to “Even though..”
Other comments:
· Line 10: “diseaseS of the colon ARE” or plural: Changed
· Line 199: define abbreviation at first mention: section removed
· Line 231: first author is Pimentel, not Lembo: Changed
· Line 231: “intestinal microbiota IN IBS to” :Changed
Reviewer 2 Report
Noha Nasef and Sunali Mehta submitted a review on the role of inflammation in pathophysiology in colonic disease: an update. The manuscript is well written and easy to read. Nevertheless, it is difficult to find “the update” as indicated in the title. For instance, authors wrote: “Recently, protease activated receptor-2 (PAR2), a g-protein coupled receptor in the GI tract that plays an important physiological role in pain and inflammation signaling [59] was linked 217 with IBS [60, 61]”, the implication of PAR2 has been already described in 2007. The reference 61 shows for the first time the endosomal activation of PAR2 in IBS not the implication of PAR2 in IBS. For the parts on which I am a little more specialist, IBS, there is a lot of imprecise sentences. For instance, based on the Rome IV criteria, there is 4 IBS subtype not 3 as indicated in this review. There is a lack of important references such as the meta-analyses performed by Bashashati et al. (Neurogastroenterology & Motility. 2017) on immune cells in IBS. The review is too superficial, authors should limit their review to one or two pathologies and add a paragraph on “the update”.
Author Response
In line with reviewer 1 and 2’s comments, we have removed the IBS section, shortened it and added as an IBD subsection (section 3.3). We have included recent study from 2020, discriminating IBS from IBD and expanded on these points in section 3.3 (lines 319 - 327). Now the review only covers 3 pathologies in details:
Diverticular disease, Inflammatory Bowel Disease and Colorectal cancer.
The aim of the manuscript was to highlight the latest research in each of the pathologies as an update. In line with this, majority of the work cited in the review is less than 5 years old (2015 – 2020: 102/160 publications, 5 – 10years old: 43/160 and only 9 references <10 years old).
Reviewer 3 Report
A very complete work was done, however, there is a lot of information related to inflammatory processes, markers, enzymes, etc. I suggest strengthening the rationale for the review.
Author Response
We would like to thank the reviewer for their complementary feedback on the review. As suggested by the reviewer we have strengthen the rationale for this work and have included it in the Introduction section (Section 1, Line 42 – 58).
Reviewer 4 Report
In the manuscript entitled "Role of inflammation in pathophysiology in colonic disease: an update", Noha Nasef. et al reported four chronic inflammation (DD, IBS, IBD, CRCs), which were common colonic disease in clinical. The authors provided an update on the role of inflammation influenced by the host immune response and microbiota in four types of colonic pathologies. This review is well presented and the quality is adequated.
Comments:
- All bacterial names need to be italicized, please modify. For example, lines 112 and 121.
- The paragraph "3.2. Influence of the microbiota in inflammation associated with IBS", the authors goes to great length on diet, I think this paragraph should focus more on microbiota.
- Line 459-461, the author mentioned "These features result in invasion of bacteria and bacterial degradation products into ....",it is better to detail what kinds of bacteria.
Author Response
We would like to thank the reviewer for their complementary and constructive feedback on the review.
1. All bacterial names need to be italicized, please modify. For example, lines 112 and 121. Done
2. The paragraph "3.2. Influence of the microbiota in inflammation associated with IBS", the authors goes to great length on diet, I think this paragraph should focus more on microbiota. We have now removed the IBS section, shortened it and added as an IBD subsection (section 3.3).
3. Line 459-461, the author mentioned "These features result in invasion of bacteria and bacterial degradation products into ....",it is better to detail what kinds of bacteria. The type of bacteria are now included in the sentence.
Round 2
Reviewer 1 Report
Thank you for the revision, I have no further comments.
Author Response
Thank you for your constructive feedback. We believe it has improved the manuscript.
Reviewer 2 Report
I have no further comments
Author Response

(The authors gave the same response as above.)
